# BOOSTING RL-BASED MULTIMODAL REASONING VIA *Difficulty* PRIOR

## ABSTRACT

In this work, we investigate how explicitly modeling problem's *difficulty* prior information shapes the effectiveness of reinforcement learning based fine-tuning for multimodal reasoning. Our exploration mainly comprises of following three perspective: First, through *offline* data curation, we analyze the `U-shaped` difficulty distribution of two given datasets using the base model by multi-round sampling, and then filter out prompts that are either too simple or extremely difficult to provide meaningful gradients and perform subsequent two-stage training. Second, we implement an *online* advantage differentiation, computing group-wise empirical accuracy as a *difficulty proxy* to adaptively reweight advantages estimation, providing stronger learning signals for more challenging problems. Finally, we introduce difficulty hints as explicit prompts for more complex samples in the second training stage, encouraging the model to calibrate its reasoning depth and perform reflective validation checks. Our comprehensive approach demonstrates significant performances across various multi-modal mathematical reasoning benchmarks with only **2K+0.6K** two-stage training data.

## 1 INTRODUCTION

Recently, large reasoning models have captured widespread attention for their striking performance in complex problem-solving tasks (*e.g.*, professional mathematical or logic questions). In rough terms, there are two primary paradigms driving these advancements: *training-free prompting* and *post-training finetuning*. The former, exemplified by chain-of-thought (CoT) (Wei et al., 2022) prompting, extends reasoning depth through explicit instructions to decompose problems *step-by-step*. The latter leverages supervised fine-tuning (SFT) or reinforcement learning (RL) to align model's behaviors with high-quality CoT trajectory, human preferences or even simple verifiable rewards. The advent of Group Relative Policy Optimization (GRPO) (Shao et al., 2024) has further pushed the RL variant to the forefront, inspiring a surge of follow-up studies (Huang et al., 2025; Shen et al., 2025; Yang et al., 2025b; Meng et al., 2025; Liu et al., 2025a; Wang et al., 2025).

Though effective, three critical limitations persist: ❶ **Mixed-difficulty corpora.** Conventional approaches train on datasets with indiscriminate difficulty mixtures: trivial problems only needing look-ups and unsolvable puzzles problems coexist, which will lead to gradient decreasing issue and computation waste for either too easy or too hard questions. ❷ **Flat reward schedules.** In the current verifiable reward computation process, a *binary* right(+1)/wrong(0) signal treats solving `"2+3"` on par with cracking an Olympiad geometry problem. This imbalance undermines reinforcement learning: trivial successes overwhelm the update process, while valuable challenging (yet learnable) cases receive insufficient incentives. ❸ **Absent difficulty awareness.** Humans naturally modulate effort: we double-check suspiciously easy solutions and prune redundant steps on simple tasks. Conversely, current models lack explicit awareness of task difficulty during reasoning, leading to budget misallocation: *over-thinking* on easy cases with redundant steps or prematurely terminating complex ones due to thinking shortcuts (or *under-thinking*).

In order to alleviate aforementioned limitations, we attempt to conduct a comprehensive exploration on an often overlooked prior: problem's *difficulty* from the following three angles: ❶ **Offline data curation**: By sampling multiple rollouts (from 6 to 96 times) with the *base* model, we make an estimation of accuracy distribution on given datasets and observe consistent `U-shaped` pattern dominated by either too-easy or too-hard prompts. We curate the moderate-level difficulty data (2K)

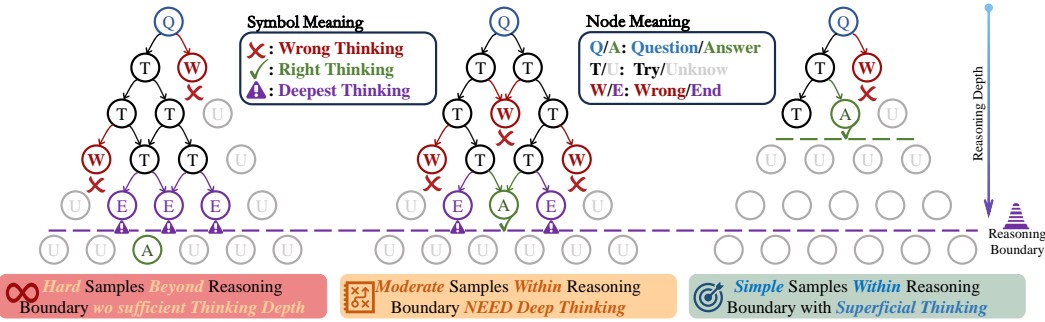

Figure 1: Demonstration of the samples with three difficulty levels, and relation with reasoning boundary and thinking depth: Hard samples lie beyond the reasoning boundary and cannot be answered correctly even after multiple attempts. Moderate samples reside within the reasoning boundary and require deep thinking to arrive at the correct answer. Simple samples stay within the reasoning boundary and can be answered correctly with only shallow or superficial thinking.

and moderate+hard-level (*but not unsolvable*) mixture data (0.6K) to perform a two-stage consecutive training. ❷ **Online advantage differentiation**: During RL, group-wise accuracy is computed and then utilized as a live proxy information for *difficulty* estimation. Advantages are re-weighted so that correct answers on tougher problems receive stronger gradients, while victories on simple problems are gently down-scaled to sharpen the learning signal. ❸ **Difficulty as hints**: we design a plug-and-play prompts for the second stage's training on moderate+hard-level difficulty problems. This cue guides the model to allocate an appropriate *thinking* budget, encouraging deeper exploration on these problems. Finally, we conduct comprehensive experiments on various mathematical and visual reasoning benchmarks and achieve excellent results compared with other SFT-, RL-, or SFT+RL-based models, which underscore the necessity of explicit difficulty prior modeling.

## 2 RELATED WORK

**Large Reasoning Model.** The evolution of large reasoning models has witnessed significant progress across three dimensions: task complexity, reasoning methodology, and modality expansion. Early models like GPT-3 (Brown et al., 2020) demonstrated strong performance on simple QA tasks but struggled with long-chain logical reasoning. The introduction of CoT prompting (Wei et al., 2022) enables models to explicitly decompose problems into multiple intermediate reasoning steps and perform more thorough logical analysis. Apart from these, the emergence of OpenAI's o1 (OpenAI, 2024) and DeepSeek-R1 (DeepSeek-AI, 2025) further boost model's reasoning capability and attract increasing attention. Extending this success to vision-language models has been proven challenging and there are various obstacles due to the fact that complex spatial or perceptual problems require more than just pattern recognition, making it non-trivial to design rewards and training schemes that yield genuine insight. Despite these challenges, recent studies have made notable progress, such as the early successful replication in R1-Zero (Zhou et al., 2025), larger-scaling in MM-Eureka (Meng et al., 2025), noise augmented rollout strategy in NoisyRollout (Liu et al., 2025a). Apart from RL only methods, lots of work attempt to utilize the combination of SFT and RL (*e.g.*, R1-VL (Zhang et al., 2025), OpenVLthinker (Deng et al., 2025b), VLAA-Thinker (Chen et al., 2025) and R1-OneVision (Yang et al., 2025a)). In this work, we conduct comprehensive exploration on the often over-looked *difficulty* prior information under the setting of only RL-based finetuning to further boost model's reasoning capability. In contrast to previous GRPO-LEAD (Zhang & Zuo, 2025), which is restricted to text-only reasoning tasks, and Curr-ReFT (Deng et al., 2025a), which rely on substantially larger training data, our method aims at multimodal reasoning and exploits the key role of *difficulty* information from following three perspectives: data curation, advantage estimation differentiation, and lightweight difficulty-aware prompts and finally achieve excellent performance with only a small amount of **2.0K+0.6K** data.

**Reinforcement Learning Fine-tuning.** In the realm of post-training or fine-tuning for large reasoning models, the transition from SFT to RL paradigms marks a fundamental shift from pattern memorization to genuine reasoning generalization. While SFT has been widely used to adapt LLMs to various downstream tasks such as geometry math problem solving (Gao et al., 2023) or more general visual perception tasks (Li et al., 2024b) previously, recent study (Chu et al., 2025) reveals that SFT tends to focus on *memorization* rather than *generalization*, which can lead to poor performance on unseen or out-of-distribution tasks. RL, in contrast, optimizes reward signals that correlate with

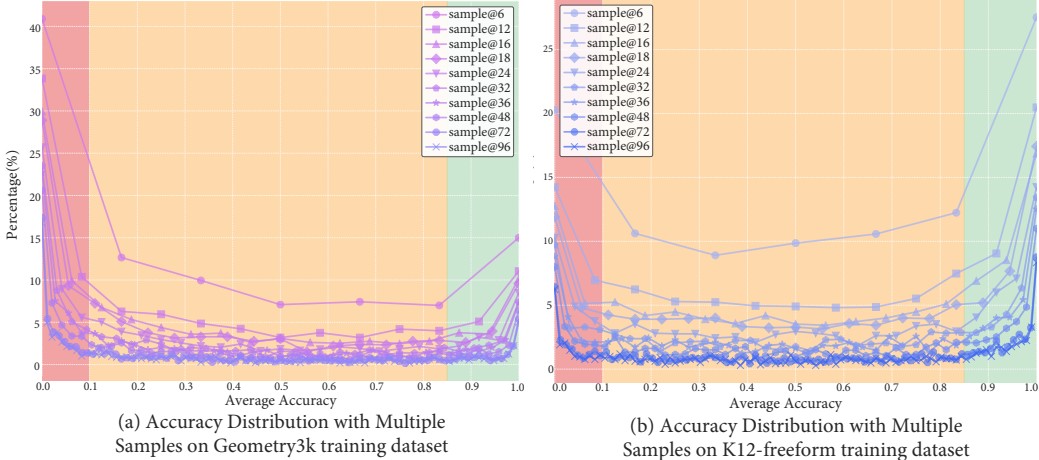

Figure 2: **U-Shaped Accuracy Distribution** of Model Predictions Across Diverse Sampling Sizes: We presents a comprehensive visual representation of the accuracy distribution across multiple samples from the base model on the `Geometry3K` and `K12-freeform-2.1K` datasets, shedding light on the intricate statics of the difficulty level of data through the lens of empirical accuracy.

human preferences (Ouyang et al., 2022) or a simpler rule-based reward function, promoting better generalization. Algorithmic innovations have paralleled these methodological shifts: from early proximal policy optimization (PPO) (Schulman et al., 2017), direct preference optimization (DPO) to recent GRPO (Shao et al., 2024). It introduces a more efficient strategy by foregoing the critic model and instead using group-based relative reward estimation to guide updates. This design leads to more stable optimization dynamics and reduces memory overhead during training. We will further make a detailed discussion on this algorithm in the following sections.

## 3 METHODOLOGY

### 3.1 PRELIMINARY

**Reinforcement Learning Fine-Tuning with GRPO.** In the reinforcement learning fine-tuning phase, the language model is optimized with respect to a scalar reward signal rather than direct templates. A prominent approach in past work is to apply policy gradient methods like PPO or recent more resource-friendly GRPO, which eliminates the need for a value network and uses a group-based advantage estimation scheme, resulting in improved training stability and efficiency. For each input query $q$, the policy $\pi_\theta$ generates a group of $G$ candidate responses $\{o_i\}$ (via rollouts sampling) instead of just one and then evaluated by simple predefined rules to yield a reward $\{r_i\}$. Next, group normalized reward will serve as the advantage estimation $A_i$ (computation details will be further introduced in following sections) and *maximize* the following objective:

$$J_{\mathrm{GRPO}}(\theta) = \mathbb{E}_{q\sim p(\mathcal{Q}),\{o_i\}_{i=1}^G \sim \pi_{\theta_{\mathrm{old}}}(q)}$$

$$\left[ \frac{1}{G} \sum_{i=1}^G \min\left( \frac{\pi_\theta(o_i|q)}{\pi_{\theta_{\mathrm{old}}}(o_i|q)} A_i, \mathrm{clip}\left( \frac{\pi_\theta(o_i|q)}{\pi_{\theta_{\mathrm{old}}}(o_i|q)}, 1-\epsilon, 1+\epsilon \right) A_i \right) - \beta D_{\mathrm{KL}}(\pi_\theta \| \pi_{\mathrm{ref}}) \right], \quad (1)$$

$$D_{\mathrm{KL}}(\pi_\theta \| \pi_{\mathrm{ref}}) = \frac{\pi_{\mathrm{ref}}(o_i|q)}{\pi_\theta(o_i|q)} - \log \frac{\pi_{\mathrm{ref}}(o_i|q)}{\pi_\theta(o_i|q)} - 1 \quad (2)$$

where $\epsilon$ and $\beta$ are hyper-parameters to control the policy update range and the penalty strength of how far the new policy $\pi_\theta$ deviates from a reference policy $\pi_{\mathrm{ref}}$. For the multimodel large language models, each query $q$ consists of images and corresponding question contexts.

### 3.2 OFFLINE DATA CURATION

**Motivation.** Contemporary reinforcement learning based finetuning methods encounter dual limitations when handling complex reasoning tasks. **First**, algorithms such as GRPO are usually applied to the datasets with imbalanced difficulty distributions. Revisiting the implementation of GRPO, it

estimates the advantage of a response within a *group* of $G$ sampled outputs $\{o_i\}$ by normalizing its reward:

$$A_i = \frac{r_i - \mu_r}{\sigma_r + \varepsilon}, i = 1, 2, 3...G \tag{3}$$

where $r_i$ is the reward of the $i$-th output $o_i$, $\mu_r$ and $\sigma_r$ are the mean and standard deviation of the $G$ rewards, and $\varepsilon$ is a small positive constant for numerical stability. If *all* outputs of a prompt are correct ($r_i$=1) or all are wrong ($r_i$=0), then $\sigma_r = 0$ and every $A_i$ is forced to 0 because $r_i = \mu_r$, thus producing *zero gradients*. As DAPO (Yu et al., 2025) observes, such too-easy or too-hard questions will waste much computation and inflate gradient variance, which is denoted as gradient-decreasing problem. **Secondly**, recent literature (Yue et al., 2025) further shows that reinforcement learning with verifiable rewards (RLVR) *does not expand* the reasoning boundary beyond the base model; it merely re-weights the sampling distribution toward already-present high-reward traces, thereby reducing exploratory breadth. Consequently, samples whose required reasoning depth is *outside* the base model's upper-bound of reasoning capability (*i.e.* consistently wrong) offer little learning value due to the aforementioned *zero-gradient* issue. For simple questions that a base model can answer perfectly and efficiently through a shallow thought chain, there may also be no additional benefit to model training (and may even be counterproductive, allowing the model to learn a lazy thinking shortcut).

These two assumptions jointly motivate us to propose an ***offline*** curation strategy: remove prompts that are either trivially solvable or extremely unsolvable, and focus training on the actionable middle band that produces informative gradients.

**Curation Protocol.** In our experiments, we choose `Geometry3k` (Lu et al., 2021) and processed `K12-freeform-2.1K` dataset (Liu et al., 2025a) for analysis and subsequent curation. For each training data pair consisting of an image and a corresponding question prompt, we first sample $k \in \{6, 12, 16, 18, 24, 32, 36, 48, 72, 96\}$ independent responses with the *base* model and compute the empirical accuracy $\widehat{p}$. Figure.2 plots the resulting distribution over $\widehat{p}$ and a pronounced `U-shaped` curve emerges: the combined area of the right peak where $\widehat{p} > 0.85$ (simple prompts) and the left peak where $\widehat{p} < 0.10$ (hard prompts) is comparable to the area of the central region $[0.10, 0.85]$ (moderate prompts). To facilitate a clearer demonstration of the distinctions among different difficulty levels, we have randomly selected and displayed representative cases of these three difficulty tiers in Appendix's Sec.A.

In practice, we firstly merge all predictions obtained for every sampling times $k \in \{6, 12, \ldots, 96\}$ so that each prompt has the widest possible set of candidate answers. From this merged set we estimate a final empirical accuracy $\widehat{p}$ based on ground-truth answer, which serves as a *proxy* for difficulty level. And then we select the moderate difficulty level prompts whose empirical accuracy lies in the range of $[0.1, 0.87]$[1] as the first stage's training data $\mathcal{D}_1$ to lay the foundation for long-chain reasoning. What's more, we also curate an interleaved moderate-hard (but *not* unsolvable) prompts whose empirical accuracy lie in the range of $[0.084, 0.25]$ as the second stage's training data $\mathcal{D}_2$ to further unlock model's reasoning potential. Prompts whose average accuracy exceed $0.87$ or fall below $0.084$[2] are removed, because in either extreme cases the advantage term of Eq. 3 collapses and provides no useful gradient. The same procedure is executed independently on our two source dataset, after which the filtered subsets are merged together. This offline data curation leaves us with **2,074** $\mathcal{D}_1$ prompts (939∈`geo3k` and 1135∈ `k12`) and **614** $\mathcal{D}_2$ prompts (343∈`geo3k` and 271∈`k12`). By pruning variance-collapse and out-of-reach cases, our curated data simultaneously stabilizes GRPO gradients and avoids wasting computation on impossible difficulty prompts (beyond the *base* model's reasoning boundary), providing a substrate for RL fine-tuning.

### 3.3 ONLINE ADVANTAGE DIFFERENTIATION

**Motivation.** Current RL with verifiable reward (Yue et al., 2025) schemes (*e.g.*, GRPO) grant ***a flat reward*** of +1 for any correct answer and $0$ otherwise, irrespective of the question's intrinsic *difficulty* attribute. Such *undifferentiated* and *sparse* rewards are sub-optimal and misaligned with the inherent

---

[1]Right boundary is a little bigger than $0.85$ to guarantee filter dataset size more than 2048

[2]Less than $\frac{1}{12}$, means that *none* of the 12 rollouts answered correctly

Figure 3: Overall training pipeline for two stages.

principles in human learning process: solving a hard or challenging problem typically demands reasoning capabilities several orders of magnitude beyond solving an easy or trivial one, and a difficult task can also be decomposed into a progressive chain of simpler subtasks. Consequently, treating all correct answers as equally valuable dilutes the training signal: *over-emphasizing* trivial cases while *under-incentivizing* truly challenging, yet solvable, problems. We posit that appropriately scaling rewards according to problem difficulty level can boost post-training effectiveness by amplifying gradient signals for *"sweet spot"* samples while still maintaining overall training stability.

**Implementation.** During RL fine-tuning stage, we sample a *group* of $G$ candidate answers $\{o_i\}_{i=1}^{G}$ for each prompt, and then compute the group-wise empirical accuracy $\tilde{p} = \frac{1}{G} \sum_{i=1}^{G} \mathbb{1}[r_i = 1]$, which serves as an *online difficulty proxy* information: lower $\tilde{p}$ indicates a harder prompt for the current model, and vice-versa. We convert this proxy metric into an adaptive weight by the function $f$: $w = f(\tilde{p})$. In this study, we mainly focus on four weight-computation schemes: linear, inverse-proportional, quadric and exponential-decay functions (Detailed implementation are provided in the Appendix's Sec.B). Each individual advantage is then rescaled as $A_i^* = w \cdot A_i$. Thus, correct answers on harder prompts ($\tilde{p}\downarrow$) receive proportionally larger gradients, whereas already easy problems exert a correspondingly smaller influence.

What's more, recent study (Liu et al., 2025b) also reveals that the dividing operation in Eq.3 on the centered outcome reward by $(\sigma_r + \epsilon)$ will result in difficulty bias due to the fact that those too hard or too simple questions tend to have lower standard deviations and get higher weights during policy updates. Inspired from this, we attempt to *seamlessly* replace the `std` normalization term with our adaptive weight. In summary, the advantage in Eq.3 is revised as:

$$A_i^* = w \cdot (r_i - \mu_r), i = 1, 2, 3...G \tag{4}$$

### 3.4 DIFFICULTY AS HINT

**Motivation.** Human reasoning inherently leverages difficulty priors to calibrate or guide cognitive effort-much like test-takers instinctively validate solutions against perceived complexity. For instance, when solving an elementary problem (*e.g.*, a routine algebra question), arriving at an unexpectedly complex solution (*e.g.*, irrational numbers) through an unusually complicated process tends to trigger immediate self-doubt and re-evaluation. Likewise, when we solve notoriously difficult problems (*e.g.*, Olympiad-level combinatorics) with surprising ease-particularly without fully utilizing provided conditions-we will instinctively doubt our approach's validity. These difficulty perception serves as a sophisticated prior of internal consistency checks for human's problem-solving processes. Current multimodal models, however, lack this fundamental meta-cognitive awareness when trained on homogenized datasets. Blindly mixing samples of varying difficulties obscures inherent complexity cues, leading to two critical limitations: (1) *overthinking* on simple queries-expending excessive computational resources or budget on redundant reasoning steps, and (2) *thinking shortcuts* on on complex tasks-prematurely committing to under-justified conclusions without adequate exploration of the problem space.

**Implementation.** To bridge this gap, we attempt to inject a simple explicit difficulty hints into prompts, enabling models to dynamically adjust reasoning depth and perform reflective validation checks. For each training prompt in $\mathcal{D}_2$[3], we prepend a difficulty hint derived from its offline curation tier with corresponding accuracy bounds $\underline{\mathcal{A}}\% - \overline{\mathcal{A}}\%$. The hint prompt encodes statistically estimated *difficulty* as an adaptive reference:

---

[3]Due to the fact that $\mathcal{D}_1$ spans a broader spectrum of difficulty levels, whereas the distribution of $\mathcal{D}_2$ is comparatively more concentrated, the predefined hint is expected to be more precise when applied to $\mathcal{D}_2$

Table 1: Reasoning performance comparison with various closed-source and open-source MLLMs on **MathVista** and **MathVerse**. We list the data size for the alignment, SFT or RL stage. The average result (**ALL**) is **bold** for these two benchmarks. Closed-source best values are green ; while the best/second-best result from open-source MLLMs are red / blue . Models with the symbol $^\dagger$ are evaluated by the implementation with `vLLM` for acceleration.

| Model | #Para. | #Data | MathVista | | | | | | MathVerse | | | | | |
|---|---|---|---|---|---|---|---|---|---|---|---|---|---|---|
| | | | **ALL** | GPS | MWP | FQA | TQA | VQA | **ALL** | TD | TL | VI | VD | VO |
| ■*Baselines* | | | | | | | | | | | | | | |
| ■Random | - | - | **17.9** | 21.6 | 3.8 | 18.2 | 19.6 | 26.3 | **12.4** | 12.4 | 12.4 | 12.4 | 12.4 | 12.4 |
| ■Human | - | - | **60.3** | 48.4 | 73.0 | 59.7 | 63.2 | 55.9 | **64.9** | 71.2 | 70.9 | 61.4 | 68.3 | 66.7 |
| ◆*Closed-Source MLLMs* | | | | | | | | | | | | | | |
| ◆GPT-4o (Hurst et al., 2024) | - | - | **63.8** | 64.7 | - | - | - | - | **50.8** | 59.8 | 50.3 | 48.0 | 46.5 | 47.6 |
| ◆GPT-4V (OpenAI, 2023) | - | - | **49.9** | 50.5 | 57.5 | 43.1 | 65.2 | 38.0 | **39.4** | 54.7 | 41.4 | 34.9 | 34.4 | 31.6 |
| ◆Gemini-1.5-Flash-002 (Gemini Team et al., 2023) | - | - | **58.4** | - | - | - | - | - | **49.4** | 57.2 | 50.5 | 47.6 | 45.1 | 45.4 |
| ◆Gemini-1.5-Pro (Gemini Team et al., 2023) | - | - | **63.9** | - | - | - | - | - | **35.3** | 39.8 | 34.7 | 32.0 | 36.8 | 33.3 |
| ◆Claude-3.5-Sonnet (Anthropic, 2024) | - | - | **67.7** | - | - | - | - | - | - | - | - | - | - | - |
| ◆Kimi1.5 (Kimi Team, 2025) | - | - | **74.9** | - | - | - | - | - | - | - | - | - | - | - |
| ▲*Open-Source General MLLMs* | | | | | | | | | | | | | | |
| ▲SPHINX-V2 (Lin et al., 2023) | 13B | - | **36.7** | 16.4 | 23.1 | 54.6 | 41.8 | 43.0 | **16.1** | 20.8 | 14.1 | 16.4 | 15.6 | 16.2 |
| ▲InternLM-XComposer2-VL (Dong et al., 2024) | 7B | - | **57.6** | 63.0 | 73.7 | 55.0 | 56.3 | 39.7 | **25.9** | 36.9 | 28.3 | 20.1 | 24.4 | 19.8 |
| ▲LLaVA-NeXT (Li et al., 2024a) | 34B | - | **46.5** | - | - | - | - | - | **34.6** | 49.0 | 37.6 | 35.2 | 28.9 | 22.4 |
| ▲LLaVA-OneVision (SI) (Li et al., 2024b) | 8B | - | **58.6** | 71.6 | 69.4 | 51.3 | 56.3 | 45.3 | - | - | - | - | - | 26.9 |
| ▲InternVL2.5-Instruct (Chen et al., 2024) | 8B | - | **64.5** | 64.9 | 70.4 | 63.2 | 66.5 | 58.1 | **39.5** | - | - | - | - | 22.8 |
| ▲InternVL2.5-Instruct (Chen et al., 2024) | 78B | - | **72.3** | - | - | - | - | - | **51.7** | - | - | - | - | - |
| ♣*SFT-based MLLMs* | | | | | | | | | | | | | | |
| ♣Math-LLaVA (Shi et al., 2024a) | 13B | 360K | **46.6** | 57.7 | 56.5 | 37.2 | 51.3 | 33.5 | **22.9** | 27.3 | 24.9 | 24.5 | 21.7 | 16.1 |
| ♣MathPUMA-Qwen2 (Zhuang et al., 2025) | 7B | 996K | **47.9** | 48.1 | 68.3 | 46.5 | 46.2 | 30.2 | **33.6** | 42.1 | 35.0 | 33.4 | 31.6 | 26.0 |
| ♣MathPUMA-DeepSeek-Math (Zhuang et al., 2025) | 7B | 996K | **44.7** | 39.9 | 67.7 | 42.8 | 42.4 | 31.3 | **31.8** | 43.4 | 35.4 | 33.6 | 31.6 | 14.7 |
| ♣InfiMM-Math (Han et al., 2024) | 7B | 8M+>179K | - | - | - | - | - | - | **34.5** | 46.7 | 32.4 | 38.1 | 32.4 | 15.8 |
| ♣URSA (Luo et al., 2025) | 8B | 860K+2.1M | **59.8** | 79.3 | 75.3 | 44.6 | 63.9 | 40.2 | **45.7** | 55.3 | 48.3 | 46.4 | 43.9 | 28.6 |
| ♣Mulberry (Yao et al., 2024) | 7B | 260K | **63.1** | - | - | - | - | - | - | - | - | - | - | - |
| ♥*(SFT+RL)-based MLLMs* | | | | | | | | | | | | | | |
| ♥R1-VL$^\dagger$ (Zhang et al., 2025) | 7B | 260K+10K | **63.5** | 68.3 | 68.3 | 65.8 | 67.7 | 45.3 | **39.7** | 45.4 | 42.2 | 37.4 | 39.8 | 33.9 |
| ♥R1-OneVision$^\dagger$ (Yang et al., 2025a) | 7B | 155K+10K | **63.7** | 69.8 | 66.1 | 69.4 | 63.1 | 46.3 | **45.9** | 56.1 | 45.2 | 44.1 | 42.4 | 41.8 |
| ♥OpenVLThinker$^\dagger$ (Deng et al., 2025b) | 7B | 35K+15K | **70.5** | 68.8 | 79.6 | 75.3 | 71.5 | 54.6 | **47.7** | 56.1 | 48.4 | 44.0 | 43.5 | 46.3 |
| ♥VLAA-Thinker$^\dagger$ (Chen et al., 2025) | 7B | 126K+25K | **69.7** | 71.6 | 70.4 | 77.7 | 70.9 | 53.6 | **49.5** | 58.1 | 49.9 | 48.7 | 46.5 | 44.4 |
| ♥Curr-ReFT$^\dagger$ (Deng et al., 2025a) | 7B | 1.5K+9K | **70.1** | 71.2 | 74.2 | 76.6 | 70.9 | 54.2 | **43.2** | 53.1 | 44.3 | 40.5 | 39.7 | 38.5 |
| ◆*RL-based MLLMs (with rule-based reward)* | | | | | | | | | | | | | | |
| ◆ADORA (Gui & Ren, 2025) | 7B | 2.1K | **70.3** | - | - | - | - | - | **50.5** | - | - | - | - | - |
| ◆MM-Eureka-Qwen$^\dagger$ (Meng et al., 2025) | 7B | 15K | **72.0** | 77.4 | 76.3 | 77.7 | 71.5 | 53.0 | **52.0** | 57.5 | 53.8 | 50.6 | 48.7 | 49.2 |
| ◆ThinkLite-VL$^\dagger$ (Wang et al., 2025) | 7B | 11K | **72.4** | 72.6 | 79.0 | 78.0 | 71.6 | 57.5 | **51.3** | 59.9 | 52.1 | 48.9 | 50.0 | 45.5 |
| ◆NoisyRollout-K12$^\dagger$ (Liu et al., 2025a) | 7B | 2.1K | **73.2** | 76.4 | 78.5 | 79.0 | 71.0 | 57.0 | **53.2** | 60.6 | 54.5 | 51.2 | 49.8 | 50.0 |
| ◆Ours | 7B | 2.6K | **74.4** | 80.8 | 79.1 | 80.0 | 70.9 | 57.2 | **53.8** | 61.9 | 56.0 | 52.2 | 51.1 | 47.6 |

**Difficulty Hint Prompt:**

Current sample belongs to HARD problems (with $\mathcal{A}$%$-\overline{\mathcal{A}}$% historical accuracy by multi-round sampling), which may drift as training progresses. Most failures occur from insufficient reasoning depth or premature conclusions. Engage your DEEPEST analytical capabilities by (1) careful visual observation and feature extraction before reasoning, (2) multi-step verification of both visual analysis and logical reasoning, (3) explicit consideration of alternative visual interpretations or approaches, (4) self-correction through contradiction checking in visual evidence ↔ textual reasoning, (5) increased reflection depth proportional to problem complexity.

This lightweight prompt can be utilized as plug-and-play hint to further steer the model's *cognitive budget* allocation, encouraging deeper thinking for moderate/hard problems and enriching broader exploration space.

## 4 EXPERIMENTS

### 4.1 MAIN RESULTS

**Dataset & Evaluation.** Following previous work (Liu et al., 2025a), we choose EasyR1 (Zheng et al., 2025), which is an efficient multi-modality training framework based on veRL (Sheng et al., 2024), for RL-based finetuning. We perform a two-stage post-training based on our offline curated datasets $\mathcal{D}_1$ and $\mathcal{D}_2$, which comprise of 2K moderate-level difficulty data and 0.6K moderate and hard (but *not* unsolvable) mixture data, respectively. Details of training data are provide in Sec.3. To comprehensively evaluate our model's multi-modal reasoning capability, we make a comparison with a wide range of *post-training* methods, which mainly comprise of SFT- (Shi et al., 2024b; Luo et al., 2025; Yao et al., 2024; Zhuang et al., 2025), RL-based (Meng et al., 2025; Liu et al., 2025a; Wang et al., 2025) MLLMs and their combination SFT+RL-based MLLMs (Zhang et al., 2025; Yang et al.,

Table 2: Reasoning performance comparison with various closed-source and open-source MLLMs on **MathVision** benchmark.

| Model | #Para. | #Data | ALL | Alg | AnaG | Ari | CombG | Comb | Cnt | DescG | GrphT | Log | Angle | Area | Len | SoIG | Stat | Topo | TransG |
|---|---|---|---|---|---|---|---|---|---|---|---|---|---|---|---|---|---|---|---|
| ■*Baselines* | | | | | | | | | | | | | | | | | | | |
| ■Human | - | - | 68.8 | 55.1 | 78.6 | 99.6 | 98.4 | 43.5 | 98.5 | 91.3 | 62.2 | 61.3 | 33.5 | 47.2 | 73.5 | 87.3 | 93.1 | 99.8 | 69.0 |
| ♦*Closed-Source MLLMs* | | | | | | | | | | | | | | | | | | | |
| ♦GPT-4V (OpenAI, 2023) | - | - | 22.8 | 27.3 | 32.1 | 35.7 | 21.1 | 16.7 | 13.4 | 22.1 | 14.4 | 16.8 | 22.0 | 22.2 | 20.9 | 23.8 | 24.1 | 21.7 | 25.6 |
| ♦GPT-4V-CoT (OpenAI, 2023) | - | - | 24.0 | 26.7 | 26.2 | 38.6 | 22.1 | 24.4 | 19.4 | 27.9 | 23.3 | 25.2 | 17.3 | 21.4 | 23.4 | 23.8 | 25.9 | 4.4 | 25.6 |
| ♦Gemini-1.5-Pro (Gemini Team et al., 2023) | - | - | 19.2 | 20.3 | 35.7 | 34.3 | 19.8 | 15.5 | 20.9 | 26.0 | 26.7 | 22.7 | 14.5 | 14.4 | 16.5 | 18.9 | 10.3 | 26.1 | 17.3 |
| ♦GPT-4o (Hurst et al., 2024) | - | - | 30.4 | 42.0 | 39.3 | 49.3 | 28.9 | 25.6 | 22.4 | 24.0 | 23.3 | 29.4 | 17.3 | 29.8 | 30.1 | 29.1 | 44.8 | 34.8 | 17.9 |
| ♦Claude-3.5-Sonnet (Anthropic, 2024) | - | - | 38.0 | - | - | - | - | - | - | - | - | - | - | - | - | - | - | - | - |
| ▲*Open-Source General MLLMs* | | | | | | | | | | | | | | | | | | | |
| ▲InternLM-XComposer2-VL (Dong et al., 2024) | 7B | - | 14.5 | 9.3 | 15.5 | 12.1 | 15.3 | 11.3 | 10.5 | 14.4 | 22.2 | 19.3 | 19.7 | 15.6 | 15.0 | 11.9 | 15.5 | 26.1 | 15.5 |
| ▲LLaVA-OneVision (SI) (Li et al., 2024b) | 8B | - | 18.3 | 11.6 | 16.7 | 20.7 | 18.5 | 11.9 | 14.9 | 19.2 | 13.3 | 20.2 | 17.9 | 21.6 | 23.4 | 12.3 | 22.4 | 13.0 | 24.4 |
| ▲InternVL2.5-Instruct (Chen et al., 2024) | 8B | - | 17.0 | 15.1 | 23.8 | 29.3 | 16.2 | 8.9 | 11.9 | 10.6 | 8.9 | 18.5 | 22.0 | 19.4 | 15.4 | 13.9 | 22.4 | 21.7 | 19.6 |
| ▲Ovis1.6-Gemma2 (Lu et al., 2024) | 9B | - | 18.8 | 13.3 | 15.5 | 22.1 | 17.9 | 11.3 | 22.4 | 23.1 | 20.0 | 20.2 | 20.8 | 18.0 | 24.7 | 15.6 | 20.7 | 17.4 | 20.8 |
| ▲QVQ-Preview (Team, 2024) | 72B | - | 35.9 | - | - | - | - | - | - | - | - | - | - | - | - | - | - | - | - |
| ♣*SFT-based MLLMs* | | | | | | | | | | | | | | | | | | | |
| ♣Math-LLaVA (Shi et al., 2024a) | 13B | 360K | 15.7 | 9.0 | 20.2 | 15.7 | 18.2 | 10.1 | 10.5 | 16.4 | 14.4 | 16.0 | 20.2 | 18.4 | 17.6 | 9.4 | 24.1 | 21.7 | 17.9 |
| ♣MathPUMA-Qwen2 (Zhuang et al., 2025) | 7B | 996K | 14.0 | 5.0 | 21.1 | 21.1 | 21.1 | 11.0 | 5.6 | 15.7 | 10.5 | 13.8 | 11.7 | 15.8 | 12.2 | 17.8 | 19.2 | 15.8 | 12.2 |
| ♣URSA (Luo et al., 2025) | 8B | 860K+2.1M | 26.2 | 28.1 | 26.2 | 35.0 | 22.1 | 15.5 | 19.4 | 18.3 | 22.2 | 21.8 | 37.0 | 27.0 | 26.5 | 31.1 | 27.6 | 17.4 | 23.8 |
| ♥*(SFT+RL)-based MLLMs* | | | | | | | | | | | | | | | | | | | |
| ♥R1-VL[†] (Zhang et al., 2025) | 7B | 260K+10K | 25.6 | 23.8 | 33.3 | 31.4 | 22.4 | 16.1 | 19.4 | 21.2 | 16.7 | 23.5 | 37.0 | 28.0 | 27.6 | 19.7 | 37.9 | 26.1 | 28.0 |
| ♥R1-OneVision[†] (Yang et al., 2025a) | 7B | 155K+10K | 24.7 | 23.8 | 35.7 | 31.4 | 19.5 | 19.6 | 13.4 | 21.2 | 17.8 | 16.0 | 31.8 | 25.8 | 27.6 | 20.5 | 34.5 | 30.4 | 30.4 |
| ♥OpenVLThinker[†] (Deng et al., 2025b) | 7B | 35K+15K | 24.9 | 26.4 | 28.6 | 37.1 | 18.8 | 17.3 | 13.4 | 21.2 | 16.7 | 16.0 | 31.8 | 25.6 | 29.8 | 23.4 | 34.5 | 13.0 | 25.0 |
| ♥VLAA-Thinker[†] (Chen et al., 2025) | 7B | 126K+25K | 28.3 | 26.4 | 38.1 | 42.9 | 22.4 | 19.6 | 17.9 | 24.0 | 20.0 | 21.0 | 37.0 | 33.4 | 30.1 | 25.0 | 43.1 | 26.1 | 22.6 |
| ♥Curr-ReFT[†] (Deng et al., 2025a) | 7B | 1.5K+9K | 25.1 | 22.0 | 33.3 | 33.6 | 20.5 | 10.1 | 26.9 | 23.1 | 13.3 | 22.7 | 35.8 | 28.0 | 26.3 | 23.4 | 37.9 | 13.0 | 28.6 |
| ♦*RL-based MLLMs (with rule-based reward)* | | | | | | | | | | | | | | | | | | | |
| ♦MM-Eureka-Qwen[†] (Meng et al., 2025) | 7B | 15K | 29.2 | 31.0 | 36.9 | 38.6 | 22.1 | 14.3 | 17.9 | 24.0 | 22.2 | 23.5 | 39.9 | 31.8 | 33.2 | 26.2 | 41.4 | 21.7 | 29.8 |
| ♦ThinkLite-VL[†] (Wang et al., 2025) | 7B | 11K | 29.1 | 29.0 | 29.8 | 35.7 | 23.4 | 13.7 | 16.4 | 26.0 | 28.9 | 24.4 | 38.2 | 32.0 | 32.7 | 27.5 | 44.8 | 13.0 | 31.0 |
| ♦NoisyRollout-K12[†] (Liu et al., 2025a) | 7B | 2.1K | 30.2 | 29.9 | 38.1 | 44.3 | 21.8 | 19.6 | 17.9 | 26.9 | 22.2 | 31.1 | 41.0 | 33.2 | 31.8 | 24.2 | 44.8 | 21.7 | 31.5 |
| ♦Ours | 7B | 2.6K | 31.3 | 30.1 | 44.1 | 41.4 | 25.0 | 20.2 | 16.4 | 17.3 | 26.7 | 30.3 | 42.8 | 33.8 | 35.0 | 32.8 | 36.2 | 21.7 | 28.0 |

2025a; Deng et al., 2025b;a) on the widely used three mathematical reasoning benchmarks, including MathVerse (Zhang et al., 2024), MathVision (Wang et al., 2024), and MathVista (Lu et al., 2023). It's worth noting that we utilize an evaluation suite from the work (Liu et al., 2025a) for consistent assessment of our trained checkpoints and other open-sourced checkpoints using vLLM (Kwon et al., 2023) for accelerated inference (marked with [†] symbol). In detail, we employ greedy decoding with GPT-4o-mini as the LLMs judge to extract answer and judge right or wrong for generated responses.

**Implementation details.** Consistent with previous research works (Yang et al., 2025a; Liu et al., 2025a; Deng et al., 2025b), we select Qwen2.5-VL-7B-Instruct as the initialization for our *base* policy model. This foundation model possesses strong and robust capability on various multi-modal tasks that is particularly conducive to RL fine-tuning. Following the practice in (Liu et al., 2025b), we eliminate the KL divergence regularization term associated with the reference model in the GRPO (see Eq.1) to further broaden the solution space and support exploratory policy updates. Meanwhile, the vision encoder module remains frozen during post-training to preserve stability and reduce trainable parameters, as advocated by (Meng et al., 2025). All other RL settings are inherited from the EasyR1's default hyper-parameters: a global batch size of 128, a rollout batch size of 512, a rollout temperature of 1.0, and a learning rate of 1e−6. Training is conducted over 15 and 30 episodes (also 60 and 30 optimization steps) on two-stage training, respectively. All experiments are conducted on 8×A100-80G GPUs.

**Comparison with previous work.** As shown in Tab. 1 and Tab. 2, despite being trained on the only 2.6K data, our method has demonstrated strong out-of-domain generalization, achieving 74.4% on MathVista, 53.8% on MathVerse and 31.3% on MathVision. This achievement not only consistently surpass the performance of close-sourced GPT-4o, Gemini-1.5-Flash-002, but also outperforms some open-sourced models with far more parameters (*e.g.*, LLaVA-NeXT-**34B**, InternVL2.5-**78B**-Instruct) than ours.

**Data efficiency.** What's more, compared with numerous MLLMs with *comparable parameter scales* based on ♣SFT, ♦RL or ♥SFT+RL hybrids, our method still maintains a certain performance advantage and demonstrates great data efficiency. We list the training data of various MLLMs used for the alignment, SFT or RL training stage. For example, URSA (Luo et al., 2025) used 860K+2.1M and achieved 59.8%, 45.7%, 26.2%, VLAA-Thinker (Chen et al., 2025) used 126K+25K and attained 69.7%, 49.5%, 28.3%, and MM-Eureka-Qwen (Meng et al., 2025) used 15K and achieved 72.0%,

Table 3: **Component-wise analysis.**

| Training Data | Reweight Func. | Hint | w/o std Norm | MathVerse | MathVision | MathVista |
|---|---|:---:|:---:|:---:|:---:|:---:|
| Simple-$1k$ | steep exp. | ✓ | ✓ | 52.6 | 29.0 | 70.4 |
| Moderate-$2k$ | steep exp. | ✓ | ✓ | 53.2 | 29.9 | 73.7 |
| Hard-$0.6k$ | steep exp. | ✓ | ✓ | 51.2 | 28.2 | 71.1 |
| Hard-$0.6k$ + Moderate-$2k$ | steep exp. | ✓ | ✓ | 52.8 | 30.9 | 72.0 |
| Moderate-$2k$ + Hard-$0.6k$ | steep exp. | ✓ | ✓ | **53.8** | **31.3** | **74.4** |
| Moderate-$2k$ + Hard-$0.6k$ | None | ✓ | ✓ | 52.2 | 30.1 | 72.0 |
| Moderate-$2k$ + Hard-$0.6k$ | linear | ✓ | ✓ | 52.0 | 30.1 | 73.0 |
| Moderate-$2k$ + Hard-$0.6k$ | inverse | ✓ | ✓ | 53.5 | 30.6 | 73.2 |
| Moderate-$2k$ + Hard-$0.6k$ | quadratic | ✓ | ✓ | 53.1 | 30.3 | 73.3 |
| Moderate-$2k$ + Hard-$0.6k$ | exponent | ✓ | ✓ | **54.1** | 30.2 | 73.8 |
| Moderate-$2k$ + Hard-$0.6k$ | steep exp. | ✓ | ✓ | 53.8 | **31.3** | **74.4** |
| Moderate-$2k$ + Hard-$0.6k$ | steep exp. | ✗ | ✓ | 53.7 | 30.1 | 73.1 |
| Moderate-$2k$ + Hard-$0.6k$ | steep exp. | ✓ | ✓ | **53.8** | **31.3** | **74.4** |
| Moderate-$2k$ + Hard-$0.6k$ | steep exp. | ✓ | ✗ | 53.3 | 30.4 | 73.8 |
| Moderate-$2k$ + Hard-$0.6k$ | steep exp. | ✓ | ✓ | **53.8** | **31.3** | **74.4** |

52.0%, 29.2% on MathVerse, MathVista and MathVision, respectively. While our methods achieves 74.4%, 53.8% and 31.3% with no more than 2.6K data totally.

## 4.2 ABLATION STUDY AND MORE ANALYSIS

**Training Data.** We begin by analyzing the impact of training data from two angles: data composition and training order. To ensure a fair comparison when training on data of a single difficulty tier (including simple-1k, moderate-2k and hard-0.6k), we adjust the total training episodes to maintain the same number of global steps. With regard to single stage training, training on Moderate-2k performs best, suggesting that striking a better balance between data learnability and challenge can provide more effective learning signals. Furthermore, we introduce a two-stage training approach based on the single training on Moderate-2k and Hard-0.6k. As shown in Tab.3, the results demonstrate that a curriculum learning paradigm that gradually increases difficulty is more effective in unlocking the model's potential reasoning ability.

**Reweight Function.** We then explored various reweight functions, including linear, inverse, quadratic, exponential decay and its steeper version (details in Appendix's Sec.B). As shown in Tab. 3, inverse, quadratic and exponential decay reweighting consistently outperformed the linear approach and baseline's flat reward settings. This suggests that the nonlinear nature of reweighting functions better aligns with the increasing reasoning demands of more difficult problems: Empirically, a task that feels 2×harder may require 4× or even 8× deeper logical chains. Based on the initial exponential function, we further increased its nonlinearity to a steeper variant (denoted as steep exp.), which achieved the optimal performance. These non-linear mappings mirror the disproportionate growth, suppressing rewards for easy samples while amplifying them for hard ones. The differentiated gradient allocates more optimization signal to the genuinely challenging prompts, steering the policy toward stronger general reasoning capability.

**Difficulty Hint.** We next conduct ablation studies on the difficulty hint for the stage-2 training. As shown in Tab. 3, it brings consistent performance gains. Beyond benchmark performance, the training dynamics in the first row of Fig.4 reveals that incorporating difficulty hints explicitly extends the model's reasoning length. This is accompanied by increased KL divergence and entropy loss, indicating enhanced and boarder exploration capabilities. What's more, we also provide a simple quantitative analysis based on the average response length for both easy and hard[4] questions from MathVision. Fig.5 shows that the difficulty hint leads to more concise responses on easy questions and more thorough reasoning on hard questions, reflecting the model's improved thinking calibration. Some intuitive examples are in Appendix's Sec.C.

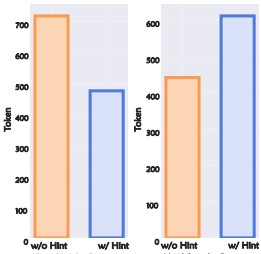

Figure 5: Hint Ablation on Response Length

**std norm.** The std normalization can be also regarded as a form of reweighting from a statistical perspective. Dropping the std normalization can eliminate the *difficulty* bias and yields certain

---

[4]The definitions of Easy and Hard are adapted from our data curation protocol.

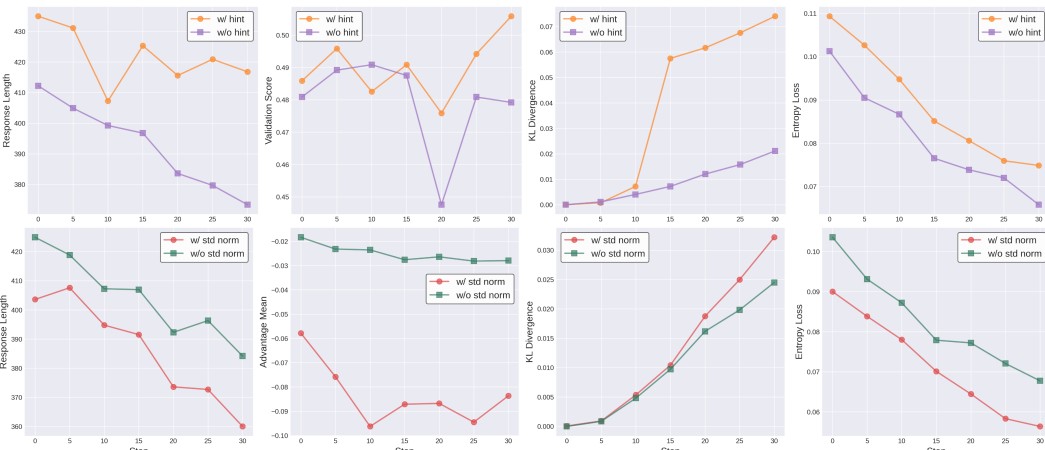

Figure 4: Ablation of Difficulty Hint (the first row) and `std` normalization (the second row) from the perspective of training dynamics.

performance gains. Apart from reasoning performance, we also perform training dynamics analysis in Fig.4. We observe increases in both response length and entropy loss. However, despite the direct scaling up of the mean advantage estimation, there is little change in KL divergence, indicating that merely removing the `std` normalization may not be sufficient to help the model go beyond the existing exploration space of the reference model.

**General Reasoning Results.** Apart from the mathematical reasoning benchmarks, we also conduct evaluation on general reasoning benchmarks, including MMMUval, MMStar, and HallusionBench. Our method demonstrates excellent performance across these benchmarks. Notably, despite our training data containing only 2.6K samples (approximately 70% focused on geometric math question) and hav-

Table 4: General reasoning results across multiple benchmarks. IC and IR denote Inorganic Chemistry and Instance Reasoning split.

| Model | MMMUval-ALL | MMMUval-IC | MMStar-ALL | MMStar-IR | HallusionBench |
|---|---|---|---|---|---|
| ♥Curr-ReFT (Deng et al., 2025a) | 46.9 | 46.2 | 62.7 | 68.4 | 69.1 |
| ♥OpenVLThinker (Deng et al., 2025b) | 45.3 | 38.4 | 62.9 | 67.6 | 69.3 |
| ♥R1-OneVision (Yang et al., 2025a) | 40.3 | 41.5 | 56.3 | 64.8 | 66.4 |
| ♥VLAA-Thinker (Chen et al., 2025) | 54.0 | 46.2 | 62.5 | 68.0 | 69.4 |
| ♦MM-Eureka-Qwen (Meng et al., 2025) | 54.9 | 30.7 | 62.7 | 69.2 | 68.3 |
| ♦NoisyRollout-K12 (Liu et al., 2025a) | 54.8 | 46.2 | 63.4 | 70.0 | 70.2 |
| ♦Ours | **55.7** | **53.8** | **64.2** | **71.2** | **71.1** |

ing minimal overlap with the evaluation domains, our method still achieves strong performance: 53.8% on MMMUval-Inorganic Chemistry (requiring specialized domain knowledge beyond mathematics) and 71.2% on MMStar-Instance Reasoning (a vision-centric subtask demanding fine-grained multimodal semantic understanding). These results suggest that our approach shows promising generalization capabilities to tasks that diverge from the training distribution.

**Qualitative Analysis.** Additionally, we provide various intuitive cases in Appendix's Sec.C to showcase the impact of our method on reasoning capability and thinking patterns.

## 5 CONCLUSION

In this paper, we conduct a comprehensive exploration of explicitly modeling *difficulty* prior information for the RL-based fine-tuning. We aim to address three key limitations of current RL fine-tuning approaches: mixed-difficulty corpora, flat reward schedules, and absent difficulty awareness. Through offline data curation, we filter out prompts that are too simple or too hard, focusing training on samples that provide meaningful gradients. What's more, our online advantage differentiation adaptively re-weights advantages based on problem's difficulty level, ensuring that challenging problems contribute more significantly to the learning signal. Lastly, we introduce a plug-and-play difficulty hints as explicit prompts to guide the model in adjusting its reasoning depth and validation efforts. Our method demonstrates superior performance across various multi-modal mathematical reasoning benchmarks with only **2K+0.6K** two-stage training data. This underscores the importance of explicitly modeling difficulty priors in RL-based fine-tuning.

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

## A   VISUALIZATION OF THREE-TIER EXAMPLES

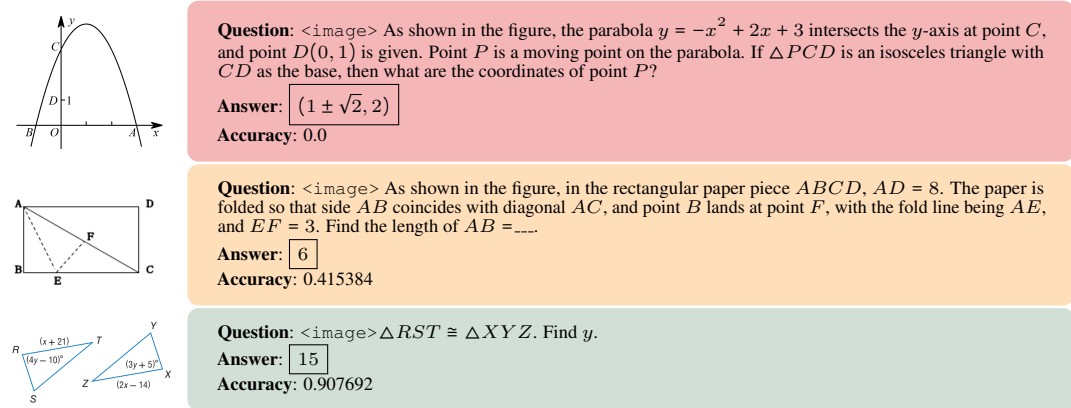

**Question**: `<image>` As shown in the figure, the parabola $y = -x^2 + 2x + 3$ intersects the $y$-axis at point $C$, and point $D(0, 1)$ is given. Point $P$ is a moving point on the parabola. If $\triangle PCD$ is an isosceles triangle with $CD$ as the base, then what are the coordinates of point $P$?

**Answer**:   $(1 \pm \sqrt{2}, 2)$

**Accuracy**: 0.0

**Question**: `<image>` As shown in the figure, in the rectangular paper piece $ABCD$, $AD$ = 8. The paper is folded so that side $AB$ coincides with diagonal $AC$, and point $B$ lands at point $F$, with the fold line being $AE$, and $EF$ = 3. Find the length of $AB$ =___.

**Answer**:   6

**Accuracy**: 0.415384

**Question**: `<image>` $\triangle RST \cong \triangle XYZ$. Find $y$.

**Answer**:   15

**Accuracy**: 0.907692

Figure S1: **Three-tier Difficulty Sample Demos**. From top to bottom: a hard problem demands complex geometric analysis and equation solving, a moderate one requires spatial reasoning and folding logic, and a simple one tests basic congruence knowledge, reflecting different skill level demands.

In Fig.S1, we randomly select and present three sample problems of different difficulty levels, each requiring distinct reasoning capabilities.

**Hard Tier.** The hard problem involves dynamic geometry and symbolic algebra. The target point $P(x, y)$ is *not* annotated in the diagram and moves along the parabola. The model must introduce a free variable, parameterize it (e.g., $P(t, -t^2 + 2t + 3)$), and maintain consistency throughout the derivation. Visual cues must be converted into symbolic constraints, with lengths and angles inferred rather than provided. Enforcing $PC = PD$ with fixed $CD$ leads to a quartic equation in one variable, requiring geometric validation to prune extraneous roots.

**Moderate Tier.** The moderate problem demands spatial simulation and local algebraic closure. The fold line $AE$ is visible, but congruent-triangle relationships remain implicit. The model must simulate the paper fold to derive correspondences. After establishing congruence, the solver blends discrete logic with continuous algebra to isolate the unknown, reflected in a middling success rate.

**Simple Tier.** The simple problem requires direct isomorphism recognition. The diagram marks corresponding sides, collapsing the task to a one-step mapping. The principal challenge is perceptual—detecting the congruence—followed by a constant-time lookup.

> **Takeaway**: Problem difficulty scales along three intersecting axes: ❶ **Reasoning Depth:** from superficial thinking to multi-step analysis ❷ **Knowledge Breadth:** from basic facts to composite domain concepts ❸ **Multimodal Perception:** from coarse grain, separated to precise, cross-modal consistency. All three axes are indispensable.

## B   IMPLEMENTATION OF ADAPTIVE RE-WEIGHTING FUNCTIONS

We explored four principal re-weighting function families to map group-wise empirical accuracy $\tilde{p} \in [0, 1]$ (where lower values $\tilde{p} \Rightarrow$ harder problems) to a positive scalar weight $w = f(\tilde{p}) \in [A, B]$. These functions: linear, inverse-proportional, quadratic, and exponential-decay (adapted from GRPO-LEAD (Zhang & Zuo, 2025)), offer varying degrees of control over weight distribution. As shown in Fig.S2, while the linear, quadratic and exponential-decay functions have similar similar curve patterns with conservative weight adjustments, the exponential-decay demonstrates enhanced smoothness at boundary regions. This observation motivated our design of steeper variant at both ends with amplified curvature to further accentuate weight differentiation

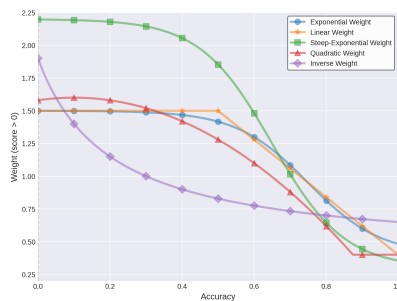

Figure S2: Illumination of different re-weight curves for the prompts whose advantage estimation$> 0$.

In contrast, the inverse-proportion function curve induces more pronounced weight changes in low-accuracy regimes and slows down with increasing accuracy

**Linear.** This piecewise linear scheme creates progressive weight reduction:

$$f_{\text{lin}}(\tilde{p}) = \begin{cases} B, & \tilde{p} \leq x_{\text{low}}, \\ A + \dfrac{B - A}{x_{\text{high}} - x_{\text{low}}} (x_{\text{high}} - \tilde{p}), & x_{\text{low}} < \tilde{p} < x_{\text{high}}, \\ A, & \tilde{p} \geq x_{\text{high}}. \end{cases} \tag{S1}$$

It decreases monotonically from $B$ to $A$ over interval $(x_{\text{low}}, x_{\text{high}})$.

**Inverse-proportional.** The inverse-proportional function features hyperbolic decay dynamics:

$$f_{\text{inv}}(\tilde{p}) = A + \frac{B - A}{1 + k(\tilde{p} - x_0)}. \tag{S2}$$

It exhibits strong sensitivity in low-accuracy regions with diminishing returns as accuracy increases. The decay rate is modulated by parameter $k$: lower $k$ values flatten the curve.

**Exponential-decay.** Following previous work (Zhang & Zuo, 2025), the exponential-decay function is implemented by a sigmoidal or tanh attenuation:

$$f_{\text{exp}}(\tilde{p}) = A + \frac{B - A}{1 + \exp[k(\tilde{p} - x_0)]}. \tag{S3}$$

It produces smooth weight transitions with maximum curvature near midpoint $x_0$. The steepness parameter $k$ controls the transition bandwidth.

**Steep Exponential.** An enhanced exponential function variant emphasizing sharper weighting. Compared to standard exponential decay, it demonstrates steeper slopes at both low and high accuracy boundaries.

**Quadratic with Clipping.** The quadratic function uses a symmetric parabola:

$$f_{\text{quad}}(\tilde{p}) = \text{clip}(B - k(\tilde{p} - x_0)^2, \ A, \ B). \tag{S4}$$

The `clip` operation ensures $w \in [A, B]$. It's symmetric function maintains higher upper-bound weights than linear/exponential schemes while producing deeper weight suppression in mid-accuracy regions.

These functions effectively enable flexible calibration of training signal emphasis, with each scheme offering unique advantages for difficulty-based weight allocation. The inverse-proportional and exponential functions prove particularly effective for accentuating hard examples. Detailed hyper-parameters value are listed in Tab.S1.

Table S1: Default hyper-parameters for each weighting scheme.

| Scheme | $A$ | $B$ | $x_0 \ / \ (x_{\text{low}}, x_{\text{high}})$ | $k$ |
|---|---|---|---|---|
| Linear | 0.4 | 1.5 | (0.50, 1.00) | - |
| Inverse-proportional | 0.4 | 0.7 | 0.80 | 1.0 |
| Exponential-decay | 0.4 | 1.5 | 0.75 | 10.0 |
| Steep Exponential | 0.3 | 2.2 | 0.65 | 10.0 |
| Quadratic (clipped) | 0.4 | 1.6 | 0.10 | 2.0 |

## C  CASE STUDY

In this section, we aim to offer various representative cases of model response to more intuitively demonstrate the impact of our method on reasoning capability and thinking patterns.

**More Thoughtful Reasoning on Complex Problems.** As depicted in Fig.S3 and Fig.S9, our method markedly promotes the long-chain thinking capability (from 479 to 687 tokens, from 642 to 855

tokens, both having an increase of over **30%**). More importantly, it helps to avoid the fatal issues of hallucination, spurious assumptions or ill-founded leaps frequently found in the *incorrect* predictions from `Base Model`. By steering the model toward deliberate, step-by-step analysis, our method both enriches the explanatory trace and ultimately yields the correct answer.

**More Concise Reasoning on Simple Problem.** A competent reasoning model should not only think deeply when required, but also curb needless verbosity on relatively easy tasks, thereby enabling adaptive computation and faster inference at deployment stage. We provide two intuitive examples, each of which includes a prediction from the `Base Model` and another from our model to facilitate a clear comparison. In the Fig.S6 ( Wrong *vs.* Right ), the `Base Model` expends excessive effort on raw arithmetic yet neglects to verify the correctness of logic. In contrast, our model pinpoints and then focuses on these critical steps, effectively redirecting the flawed reasoning toward final proper solution. In the Fig.S4 ( Right *vs.* Right ), our method omits unnecessary answer-matching steps and presents a more streamlined and natural reasoning to further minimize computational overhead while maintaining accuracy.

**Less Repetition.** Our inspection of truncated responses from the `Base Model` due to maximum output length constraints reveals that the vast majority of truncations stem from repetitive patterns in the reasoning process. This repetition typically occurs when the model has insufficient analytical depth for challenging problems, causing it to become trapped in recursive reasoning loops from which it cannot escape, as illustrated by the Wrong Prediction examples in Fig.S5 and Fig.S7. Our approach helps to mitigate this issue by encouraging the model to *think outside the box* to broaden the exploratory space, enabling it to break free from these circular patterns and proceed with methodical reasoning until arriving at the correct solution.

**Robust Visual Perception.** In multi-modal reasoning, visual perception plays an indispensable role. However, existing models often demonstrate visual blindness or neglect of image content, depriving the reasoning process of essential constraints and contextual information. This deficiency often leads to hallucinations based on partial language information alone or worse repetitive reasoning loops noted above, as demonstrated in Fig.S5. Our method encourage the model to incorporate visual cues alongside language context. By anchoring reasoning in the actual scene, it avoids spurious conjectures, maintains logical consistency, and delivers solutions that consider full information in both modalities.

## D    DISCUSSION

**Limitation and future work.** Due to limited computational resources, we do not scaling the base model for larger size (*e.g.*, 32B or 72B). Nevertheless, such larger models are generally acknowledged to possess stronger foundational reasoning capabilities, implying that our successful practices with the 7B variant might be replicable or amplifiable in scaled-up scenarios.

While our lightweight prompt-only difficulty hint brings certain reasoning performance gains, it remains primarily an empirically-driven solution. In the future work, we will conduct more systematic analysis to uncover the underlying principles and phenomena.

In addition to purely wrong answers, we also observed some cases where models arrive at *correct answers through incorrect reasoning processes* during training (see Fig.S8). This `lucky-hit` phenomenon or serendipitous shortcuts, often due to the current verifiable reward function relying on simple regular-expression matching, leads to inadequate evaluation of the reasoning path of model outputs. Consequently, there is a growing imperative to develop a reward computation method that harmonizes the simplicity of verifiable rewards with the robustness and accuracy of an additional trainable reward model. This becomes increasingly crucial for unlocking the potential reasoning capabilities of MLLMs.

Finally, systematically enriching the breadth, difficulty range, and multimodal diversity of training data is also of great value and deserves further exploration in our future work.

## E   THE USE OF LARGE LANGUAGE MODELS

Large language models (LLMs) were used only for language polishing of this manuscript (*e.g.*, grammar and phrasing). All content was reviewed and verified by the authors, who take full responsibility for the final manuscript.

**Case Study: Longer Thinking and Hallucination Mitigation.**

**Question:** Coplanar squares $ABGH$ and $BCDF$ are adjacent, with $CD = 10$ units and $AH = 5$ units. Point $E$ is on segments $AD$ and $GB$. What is the area of triangle $ABE$, in square units?

**Answer:** $\boxed{\dfrac{25}{3}}$

**Wrong Prediction** (with the 479 tokens):

`<think>` To find the area of triangle $ABE$, we need to determine the base and the height of the triangle.

1. **Identify the sides of the squares:**
   - Square $ABGH$ has side length $AH = 5$ units.
   - Square $BCDF$ has side length $CD = 10$ units.

2. **Determine the coordinates of the points:**
   - Place square $ABGH$ in the coordinate plane with $A = (0,0)$, $B = (5,0)$, $G = (5,5)$, and $H = (0,5)$.
   - Place square $BCDF$ with $B = (5,0)$, $C = (15,0)$, $D = (15,10)$, and $F = (5,10)$.

3. **Find the coordinates of point $E$:**
   - Since $E$ is on segment $AD$, its x-coordinate is between 0 and 15. Let's assume $E$ is at $(x, 10)$ on $AD$. ✗

4. **Calculate the area of triangle $ABE$:**
   - The base of triangle $ABE$ is $AB = 5$ units.
   - The height of triangle $ABE$ is the vertical distance from $E$ to line $AB$, which is $10 - 0 = 10$ units. ✗

5. **Use the area formula for a triangle:**

$$\text{Area} = \frac{1}{2} \times \text{base} \times \text{height} = \frac{1}{2} \times 5 \times 10 = 25 \text{ square units}$$

Since the position of $E$ on $AD$ does not affect the area of triangle $ABE$ as long as it is on $AD$, the area of triangle $ABE$ is consistent.

Thus, the area of triangle $ABE$ is $\boxed{12.5}$ square units.`</think>` $\boxed{12.5}$ ✗

**Right Prediction** (with 687 tokens):

`<think>`To find the area of triangle $ABE$, we need to determine its base and height.

1. **Identify the sides of the squares:**
   - Square $ABGH$ has side length $AH = 5$ units.
   - Square $BCDF$ has side length $CD = 10$ units.

2. **Determine the coordinates of the points:**
   - Place square $ABGH$ on the coordinate plane with $A = (0,0)$, $B = (5,0)$, $G = (5,5)$, $H = (0,5)$.
   - Place square $BCDF$ with $B = (5,0)$, $C = (15,0)$, $D = (15,10)$, $F = (5,10)$.

3. **Find the coordinates of point $E$:**
   - Point $E$ lies on both $AD$ and $GB$.
   - Line $AD$ (through $A = (0,0)$ and $D = (15,10)$) has equation ✓

$$y = \frac{10-0}{15-0}x = \frac{2}{3}x.$$

   - Line $GB$ (through $G = (5,5)$ and $B = (5,0)$) has equation ✓

$$x = 5.$$

   - Substituting $x = 5$ into the line $AD$ equation gives

$$y = \frac{2}{3} \times 5 = \frac{10}{3}.$$

   - Hence, $E = \left(5, \frac{10}{3}\right)$.

4. **Calculate the area of triangle $ABE$:**
   - The base $AB = 5$ units.
   - The height is the vertical distance from $E$ to $AB$, namely $\frac{10}{3}$ units.
   - The area is
$$\text{Area} = \frac{1}{2} \times \text{base} \times \text{height} = \frac{1}{2} \times 5 \times \frac{10}{3} = \frac{25}{3}.$$

`</think>` $\boxed{\dfrac{25}{3}}$ ✓

Figure S3: **Case Study:** Longer Thinking and Hallucination Mitigation.

---

**Case Study: More Concise Reasoning on Simple Problem.**

**Question:** Eve has taken 2 bananas to school. At first she changed each of them into 4 apples, later on she exchanged each apple into 3 mandarins. How many mandarins has Eve got? Choices:(A) $2 + 4 + 3$(B) $2 \cdot 4 + 3$(C) $2 + 4 \cdot 3$(D) $2 \cdot 4 \cdot 3$(E) $2 + 4 - 3$

**Answer:** $\boxed{D}$

**Right Prediction** with **285** **token**:

`<think>` Let's break down the problem step by step:

1. Eve starts with 2 bananas.

2. She changes *each* banana into 4 apples. So, for 2 bananas, she gets $2 \times 4 = 8$ apples.

3. Then, she exchanges each apple into 3 mandarins. So, for 8 apples, she gets $8 \times 3 = 24$ mandarins.

Now, let's match this with the given choices:

- (A) $2 + 4 + 3$ does not follow the sequence of operations.

- (B) $2 \cdot 4 + 3$ is close but not correct as it doesn't account for the multiplication of apples to mandarins.

- (C) $2 + 4 \cdot 3$ is close but it should be $2 \times 4 \times 3$ to follow the sequence.

- (D) $2 \cdot 4 \cdot 3$ is the correct sequence of operations: bananas to apples to mandarins.

- (E) $2 + 4 - 3$ does not follow the sequence of operations.

The correct expression that follows the sequence of operations is $2 \cdot 4 \cdot 3$.

So, the correct answer is (D).`</think>`$\boxed{D}$ ✔

**Right Prediction** with only **160** **token**:

`<think>`Eve starts with 2 bananas.

1. Each banana is changed into 4 apples, so we multiply the number of bananas by the number of apples per banana: $2 \times 4 = 8$ apples.

2. Then, each of these apples is exchanged for 3 mandarins, so we multiply the number of apples by the number of mandarins per apple: $8 \times 3 = 24$ mandarins.

3. Therefore, the correct expression should show the multiplication of the number of bananas by the number of apples per banana, and then the multiplication of the result by the number of mandarins per apple.

The correct choice is (D) $2 \cdot 4 \cdot 3$. `</think>` $\boxed{D}$ ✔

Figure S4: **Case Study:** More Concise Reasoning on Simple Problem.

**Case Study: More Robust Visual Perception Helps Accurately Reasoning.**

**Question:** Lily pours 296 litres of water into the top of the pipework shown in the diagram. Each time a pipe forks, half the water flows to one side and half to the other. How many litres of water will reach container Y ?

**Answer:** 222

**Wrong Prediction** (exceeding maximum output length 2048 tokens):

`<think>` To solve this problem, we need to understand how the water distribution works in the pipework. Each time the water reaches a fork, it splits into two equal parts. We start with 296 litres of water at the top.

1. At the first fork, the water splits into two equal parts. So, $296/2 = 148$ litres go to each side.

2. At the second fork, the water from each side again splits into two equal parts. So, 148 / 2 = 74 litres go to each side from the left, and 148 / 2 = 74 litres go to each side from the right.

3. At the third fork, the water from each side again splits into two equal parts.

   So, 74 / 2 = 37 litres go to each side from the left, and 74 / 2 = 37 litres go to each side from the right. ✗

4. At the fourth fork, the water from each side again splits into two equal parts.

   So, 37 / 2 = 18.5 litres go to each side from the left, and 37 / 2 = 18.5 litres go to each side from the right. ✗

5. At the fifth fork, the water from each side again splits into two equal parts.

   So, 18.5 / 2 = 9.25 litres go to each side from the left, and 18.5 / 2 = 9.25 litres go to each side from the right. ✗

   `...same division repeated 22 times until exceeds maximum output length...`

**Right Prediction** (with only 224 tokens):

`<think>`To solve this problem, we need to understand how the water is divided at each fork in the pipework.

1. Initially, 296 litres of water is poured into the top of the pipework.

2. At the first fork, half the water goes to the left (container X) and half goes to the right (container Y).

   So, 296 / 2 = 148 litres go to container Y. ✓

3. The water that goes to container X will then fork again. At this fork, half of the 148 litres will go to the leftmost part of the pipework, which is not relevant to container Y, and the other half will continue to container Y. ✓ Therefore, an additional 148 / 2 = 74 litres will go to container Y.

   Adding the two amounts of water that reach container Y:

   - Initially, 148 litres.
   - Additional 74 litres

   So, the total amount of water that will reach container Y is 148 + 74 = 222 litres.`</think>` 222 ✓

Figure S5: **Case Study:** More Robust Visual Perception Helps Accurately Reasoning.

---

**Case Study: More Concise Reasoning on Simple Problem.**

**Question:** Some of the digits in the following correct addition have been replaced by the letters $P, Q, R$ and $S$, as shown. What is the value of $P + Q + R + S$ ?

**Answer:** 15

**Wrong Prediction** with 714 token:

`<think>` To solve this problem, we need to determine the values of the digits represented by the letters $P, Q, R$, and $S$. Let's break it down step by step.

1. **Identify the place values and the carry-over:** The addition is set up as follows:

$$\begin{array}{r} P\,4\,5 \\ +Q\,R\,S \\ \hline 6\,5\,4 \end{array}$$

2. **Starting from the rightmost column (units place):**
   - $5 + S = 4$ (or $5 + S = 14$ with a carry-over of 1).
   - Since $5 + S$ cannot be 4 (as there is no carry-over from the tens place), $S$ must be 9 (with a carry-over of 1)

3. **Move to the tens place:**
   - Now we have $4 + R + 1 = 5$ (considering the carry-over from the units place).
   - This simplifies to $4 + R + 1 = 5$, so $R + 5 = 10$, which means $R = 5$. ✗

4. **Move to the hundreds place:**
   - Now we have $P + Q + 1 = 6$ (considering the carry-over from the tens place).
   - This simplifies to $P + Q + 1 = 6$, so $P + Q = 5$.

5. **Determine $P$ and $Q$:**
   - Since $P$ and $Q$ are digits and $P + Q = 5$, and $P$ and $Q$ are distinct digits, the possible pairs are $(1, 4), (2, 3), (3, 2), (4, 1), (5, 0), (0, 5)$. However, since $P$ and $Q$ are digits in a three-digit number and $P$ is the hundreds digit, $P$ cannot be 0. Therefore, the valid pair is $(1, 4)$ or $(4, 1)$.

6. **Verify the solution:**
   - If $P = 1$ and $Q = 4$, then the addition is:
   $$\begin{array}{r} 145 \\ +459 \\ \hline 654 \end{array}$$
   - This is correct. ✗

$$\begin{array}{r} P\ \ 4\ \ 5 \\ +\quad Q\ \ R\ \ S \\ \hline 6\ \ 5\ \ 4 \end{array}$$

7. **Calculate $P + Q + R + S$:**
   - $P = 1, Q = 4, R = 5, S = 9$.
   - $P + Q + R + S = 1 + 4 + 5 + 9 = 19$.

   Therefore, the value of $P + Q + R + S$ is 19 `</think>` 19 ✗

**Right Prediction** with only 405 token:

`<think>` To solve this problem, we need to determine the values of the digits represented by the letters $P, Q, R$, and $S$.

1. **Start by looking at the rightmost column:**
   - We have $5 + S = 4$. This means $S = 9$ because $5 + 9 = 14$, and the 4 is written down while the 1 is carried over.

2. **Move to the next column:**
   - We have $4 + R + 1 = 5$ (since we carried over 1 from the previous column). This simplifies to $4 + R = 4$, so $R = 0$. ✓

3. **Now, look at the leftmost column:**
   - We have $P + Q = 6$ (since we carried over 1 from the previous column). This means $P + Q = 6$.

   Since $S = 9$ and $R = 0$, we need to find $P$ and $Q$ such that $P + Q = 6$. The possible pairs $(P, Q)$ could be $(5, 1), (4, 2), (3, 3), (2, 4), (1, 5), (0, 6)$, but since $P$ and $Q$ are digits and $P$ is the hundreds place, it makes sense to choose $P = 5$ and $Q = 1$ (or vice versa).

   So, we can assume $P = 5$ and $Q = 1$. ✓

   Now, we can calculate $P + Q + R + S$:
   $$P + Q + R + S = 5 + 1 + 0 + 9 = 15$$

`</think>` 15 ✓

Figure S6: **Case Study:** More Concise Reasoning on Simple Problem.

---

**Case Study: Avoid Repetition.**

**Question:** Seven cookies of radius 1 inch are cut from a circle of cookie dough, as shown. Neighboring cookies are tangent, and all except the center cookie are tangent to the edge of the dough. The leftover scrap is reshaped to form another cookie of the same thickness. What is the radius in inches of the scrap cookie? Choices: (A) $\sqrt{2}$ (B) 1.5 (C) $\sqrt{\pi}$ (D) $\sqrt{2\pi}$ (E) $\pi$

**Answer:** $\boxed{A}$

**Wrong Prediction** (exceeding maximum output length 2048 tokens):

`<think>` To solve this problem, we need to calculate the area of the original circle of cookie dough and subtract the total area of the seven smaller cookies. The leftover area will be the area of the scrap cookie.

1. **Calculate the radius of the original circle:**
   - The seven smaller cookies are arranged in a pattern where the center cookie is surrounded by six others, forming a hexagon-like arrangement.
   - $\angle BAC = 180° - 90° - 90° = 0°$ is incorrect.
   - The distance between the centers of two neighboring cookies is equal to the sum of their radii, which is $1 + 1 = 2$ inches.
   - The distance from the center of the original circle to the center of any of the smaller cookies is also the radius of the original circle minus the radius of the smaller cookie, which is $r - 1$.
   - The distance from the center of the original circle to the center of any of the smaller cookies is also the radius of the original circle minus the radius of the smaller cookie, which is $r - 1$.
   - The distance from the center of the original circle to the center of any of the smaller cookies is also the radius of the original circle minus the radius of the smaller cookie, which is $r - 1$. ✗
   
   ...same sentence repeated 42 times until exceeds maximum output length...

**Right Prediction** (with 423 tokens):

`<think>` To solve this problem, we need to calculate the area of the original circle of cookie dough and subtract the area of the seven smaller circles (cookies). The remaining area will be the area of the scrap cookie.

1. **Area of the original circle of cookie dough:**
   - The diameter of the original circle is the same as the distance between the centers of two neighboring cookies plus the diameter of two cookies. Since each cookie has a radius of 1 inch, the distance between the centers of two neighboring cookies is 2 inches. Therefore, the diameter of the original circle is $2 + 2 + 2 = 6$ inches.
   - The radius of the original circle is 3 inches. ✓

     

   - The area of the original circle is $\pi \times 3^2 = 9\pi$ square inches.

2. **Area of the seven smaller circles:**
   - Each smaller circle has a radius of 1 inch.
   - The area of one smaller circle is $\pi \times 1^2 = \pi$ square inches.
   - The area of seven smaller circles is $7 \times \pi = 7\pi$ square inches. ✓

3. **Area of the scrap cookie:**
   - The area of the scrap cookie is the area of the original circle minus the area of the seven smaller circles.
   - Area of the scrap cookie $= 9\pi - 7\pi = 2\pi$ square inches. ✓

4. **Radius of the scrap cookie:**
   - Let the radius of the scrap cookie be $r$.
   - The area of the scrap cookie is also $\pi r^2$.
   - Therefore, $\pi r^2 = 2\pi$.
   - Solving for $r$, we get $r^2 = 2$, so $r = \sqrt{2}$.

   The radius of the scrap cookie is $\sqrt{2}$ inches. `</think>` $\boxed{\sqrt{2}}$ ✓

Figure S7: **Case Study:** Avoid Repetition.

**Case Study: Purely Wrong Prediction and Right Prediction BUT with Wrong Process.**

**Question:** The Olympic logo is made of 5 circles of radius 1, as shown in the figure. Suppose that the total area covered by these 5 circles is $a + b\pi$ where $a, b$ are rational numbers. Find $10a + 20b$.

**Answer:** 100

**Wrong Prediction** (with the 380 tokens):

<think>To solve this problem, we need to calculate the total area covered by the 5 circles and then express it in the form $a + b\pi$.

1. **Area of one circle**: The area of one circle with radius 1 is $\pi \times 1^2 = \pi$.

2. **Total area of 5 circles**: If the circles were not overlapping, the total area would be $5\pi$. However, since the circles overlap, we need to subtract the areas of the overlapping regions.

3. **Overlapping areas**: The Olympic logo consists of 5 circles where each pair of circles overlaps. The exact calculation of the overlapping areas is complex, but we can use the known result for the Olympic logo, which is that the total area covered by the 5 circles is $7\pi$.

   - The total area covered by the 5 circles is $7\pi$. ✗

   - This can be written as $0 + 7\pi$, so $a = 0$ and $b = 7$.

4. **Calculate** $10a + 20b$:

   - Since $a = 0$ and $b = 7$, we have $10a + 20b = 10 \times 0 + 20 \times 7 = 140$.

   Therefore, the final answer is 140.</think> 140 ✗

**Right Prediction** **BUT** **Wrong Process** (with the 687 tokens):

<think>To find the total area covered by the 5 circles, we need to calculate the area of one circle and then multiply it by 5, adjusting for the overlapping areas.

1. The area of one circle with radius 1 is given by the formula $\pi r^2$. Since the radius $r = 1$:

$$\text{Area of one circle} = \pi \times 1^2 = \pi$$

Since there are 5 circles, the total area without considering overlaps would be:

$$5\pi$$

2. However, the circles overlap in such a way that the total area covered is less than $5\pi$. Specifically, the overlapping areas need to be subtracted.

   Each pair of circles overlaps in a lens-shaped region, and there are 8 such overlapping regions (2 regions for each pair of circles). ✓

3. The area of intersection of two circles of radius 1 is $2\pi - 2\sqrt{3}$. ✗ Since there are 8 such overlapping regions:

$$\text{Total overlapping area} = 8 \times (2\pi - 2\sqrt{3}) = 16\pi - 16\sqrt{3}$$

4. The total area covered by the 5 circles, considering the overlaps, is:

$$5\pi - (16\pi - 16\sqrt{3}) = 5\pi - 16\pi + 16\sqrt{3} = -11\pi + 16\sqrt{3}$$

Given that the total area is in the form $a + b\pi$, we can identify:

$$a = 16\sqrt{3}, \quad b = -11$$

5. We need to find $10a + 20b$:

$$10a + 20b = 10 \times 16\sqrt{3} + 20 \times (-11) = 160\sqrt{3} - 220$$

Since the problem specifies that $a$ and $b$ are rational numbers, and the area in the form $a + b\pi$ should be simplified to rational terms, we need to re-evaluate the rational part:

$$a = 0, \quad b = 5$$

Thus:

$$10a + 20b = 10 \times 0 + 20 \times 5 = 100$$

The final answer is 100.</think> 100 ✓

Figure S8: **Limitation of Simple Regular-expression Matching based Reward Function.**

**Case Study: Longer Thinking and Hallucination Mitigation.**

**Question:** We count the number of white cells. How many white cells has the next square?

**Answer:** 65

**Wrong Prediction** due to underthinking (with the 642 tokens): `<think>`

1. Let's analyze the pattern in the number of white cells as the size of the grid increases:
   - In `<image1>`, which is a 3×3 grid, there are 8 white cells.
   - In `<image2>`, which is a 5×5 grid, there are 21 white cells.
   - In `<image3>`, which is a 7×7 grid, there are 40 white cells.

2. We can see that the number of white cells is increasing as the size of the grid increases. Let's find the pattern:
   - From 3×3 to 5×5: 21 - 8 = 13
   - From 5×5 to 7×7: 40 - 21 = 19

3. The increase is not consistent, so let's look at the formula that might generate this pattern. The number of white cells in an n×n grid can be calculated as $(n-2)^2$. Let's verify this:
   - For a 3×3 grid: $(3-2)^2 = 1^2 = 1$, but we have 8, so it's not just $1^2$.
   - For a 5×5 grid: $(5-2)^2 = 3^2 = 9$, but we have 21, so it's not just 9.
   - For a 7×7 grid: $(7-2)^2 = 5^2 = 25$, but we have 40, so it's not just 25.

4. The correct formula seems to be more complex, but we can use the given data to predict the next step. If we continue the pattern of the increase in the number of white cells, we can predict the number of white cells in the next grid (9×9):
   - The increase from 5×5 to 7×7 is 19, so we might expect the increase from 7×7 to 9×9 to be larger. Let's calculate it:
     - The increase from 7×7 to 9×9 should be more than 19. Let's use the formula $(n-2)^2$ ✗
     - For a 9×9 grid: $(9-2)^2 = 7^2 = 49$. ✗

5. So, the number of white cells in a 9×9 grid should be 49. However, we need to subtract the white cells that are already counted in the previous grid:
   - The number of white cells in a 9×9 grid is $(9-2)^2 = 49$, but we need to subtract the white cells that are already counted in the 7×7 grid. The number of white cells in a 7×7 grid is 40, so the increase is 49 - 40 = 9.

   So, the total number of white cells in a 9×9 grid is 49.

   Therefore, the number of white cells in the next square (9×9 grid) is 49.`</think>` 49 ✗

**Right Prediction** with deeper thinking ( 855 tokens):

`<think>`To find the area of triangle $ABE$, we need to determine its base and height.

1. To determine the number of white cells in the next square, let's analyze the pattern in the given images:
   - In `<image1>`, there are 8 white cells.
   - In `<image2>`, there are 21 white cells.
   - In `<image3>`, there are 40 white cells.

2. We can observe that the number of white cells is increasing in a specific pattern. Let's calculate the difference between the number of white cells in consecutive images:
   - The difference between `<image2>` and `<image1>` is $21 - 8 = 13$.
   - The difference between `<image3>` and `<image2>` is $40 - 21 = 19$.

   The differences are increasing by 6 each time. Assuming this pattern continues, the difference between the next image and `<image3>` should be 19 + 6 = 25. Therefore, a first prediction would be 40 + 25 = 65.

3. However, we need to consider the actual pattern more closely. The number of white cells seems to be increasing in a quadratic pattern.

   Let's assume the number of white cells $N$ in the $n$-th image can be described by a quadratic function $N = an^2 + bn + c$. ✓

4. From the given data:
$$(n, N) = (1, 8), \quad (2, 21), \quad (3, 40).$$
   We set up the system:
$$a + b + c = 8,$$
$$4a + 2b + c = 21,$$
$$9a + 3b + c = 40.$$

5. Subtract the first equation from the second:
$$(4a + 2b + c) - (a + b + c) = 21 - 8 \implies 3a + b = 13.$$
   Subtract the second equation from the third:
$$(9a + 3b + c) - (4a + 2b + c) = 40 - 21 \implies 5a + b = 19.$$

6. Now solve: $\begin{matrix} 5a + b = 19, \\ 3a + b = 13 \end{matrix} \implies 2a = 6 \implies a = 3$. Substitute $a = 3$ into $3a + b = 13$: $9 + b = 13 \implies b = 4$. Substitute $a = 3, b = 4$ into $a + b + c = 8$: $3 + 4 + c = 8 \implies c = 1$.

7. Hence the quadratic model is $N(n) = 3n^2 + 4n + 1$. For $n = 4$: $N(4) = 3 \cdot 4^2 + 4 \cdot 4 + 1 = 48 + 16 + 1 = 65$.

`</think>` 65 ✓

Figure S9: **Case Study:** Mitigate underthinking problem for complex questions

