# OpenReview forum: "Boosting RL-based Multimodal Reasoning via Difficulty Prior"
_ICLR.cc/2026/Conference — ICLR 2026 Conference Withdrawn Submission_

### Official Review · Reviewer_E3Fs · 2025-10-26

**Soundness:** 2
**Presentation:** 2
**Contribution:** 1
**Rating:** 4
**Confidence:** 4

**Summary:**

This paper proposes a RL method that leverages prior knowledge of problem difficulty to enhance the performance of MLLMs on mathematical reasoning tasks. The authors systematically incorporate difficulty information from three perspectives:

1. **Offline Data Filtering**: The difficulty of each sample is estimated through multiple rounds of sampling. Samples that are too easy (accuracy > 0.87) or extremely hard (accuracy < 0.084) are filtered out. The remaining data—approximately 2K medium-difficulty and 0.6K medium-to-hard samples—are retained for two-stage training.
2. **Online Advantage Reweighting**: During GRPO training, the weights of the advantage function are dynamically adjusted based on the group accuracy of sampled responses for each problem, thereby assigning stronger learning signals to harder questions.
3. **Difficulty Hint**: In the second training stage, explicit difficulty hints are added to hard samples to guide the model to allocate more reasoning resources and perform reflective verification.

**Strengths:**

1. **Clear Problem Focus**: The work points out three key issues in current RL fine-tuning—mixed-difficulty data, flat rewards, and lack of difficulty awareness—and proposes targeted solutions.
2. **Comprehensive Empirical Analysis**: The paper includes ablation studies, response-length analysis, and case studies, effectively validating the contribution of each component.
3. **Preliminary Generalization Evidence**: Strong performance is also observed on non-mathematical multimodal reasoning benchmarks such as MMMU and MMStar, suggesting the method has some degree of generalizability.

**Weaknesses:**

1. **Modest Performance Gains**: Although the method outperforms baselines, the absolute improvement is limited and sometimes marginal on certain subtasks, which may not sufficiently demonstrate its breakthrough potential.
2. **Extremely Small and Narrow Training Data**: The entire dataset consists of only 2.6K samples, primarily drawn from geometric math problems (Geometry3K and K12-freeform). This lack of diversity raises concerns about the method’s applicability to more complex or open-domain multimodal tasks.
3. **Methodologically Simple**: The three proposed techniques are essentially heuristic modifications to existing RL training pipelines, lacking theoretical depth or novel algorithmic innovation.

**Questions:**

Refer to the weaknesses above.

---

### Official Review · Reviewer_SB4P · 2025-10-28

**Soundness:** 2
**Presentation:** 2
**Contribution:** 2
**Rating:** 4
**Confidence:** 5

**Summary:**

This paper proposed a reinforcement learning fine-tuning framework that explicitly models task difficulty to enhance multimodal reasoning. The authors introduces three complementary components: (1) offline data curation that filters out too-easy or unsolvable samples based on U-shaped difficulty distribution; (2) online advantage differentiation that reweights gradients based on problem difficulty; and (3) difficulty hints embedded in prompts to calibrate reasoning depth. Experiments on multimodal reasoning benchmarks (MathVista, MathVerse, MathVision) show improved data efficiency and reasoning performance compared to existing RL- and SFT-based models.

**Strengths:**

- My understanding about this work is empirical investigation into multimodal reasoning models. It integrates difficulty priors into reinforcement learning for MLLMs, which is reasonable to address gradient collapse issues in mixed-difficulty datasets. (Even though I think difficulty prior is a recent common sense in improve RLVR efficiency)
- The experiments are realtively comprehensive in evaluating data efficiency and generalization. The strategy achieves better results with less training samples, outperforming models trained on much larger datasets.

**Weaknesses:**

- My major concern lies in the novelty of designing the difficulty metrics as a lot of recent works have researched this point [1-4]. These should be well discussed in the manuscript. Meanwhile, The difficulty estimation process relies heavily on empirical accuracy thresholds, which may not generalize well to new datasets or domains. Note that 183-192 clarifies the precise interval in Geometry3K, while I am worried about its scalability to other datasets.
- The methodological contribution overlaps with prior curriculum learning paradigms, and the distinction between “difficulty prior” and traditional curriculum learning could be made clearer.
- I am wondering the application scenario of this work. Note that multimodal reasoning is not well developed as unimodal reasoning models. Are there any special design for multimodal scenarios? Can the adopted strategy work in LLMs?


Other Minor Questions:

- a) How sensitive is model performance to the exact accuracy thresholds used for data curation (e.g., [0.1, 0.87])?
- b) Can the proposed difficulty reweighting and hint mechanisms be generalized beyond mathematical reasoning to other domains (e.g., scientific or commonsense reasoning)?
- c) Does the difficulty hint risk introducing label leakage or biasing the model’s reasoning path artificially?

References:

[1] Single-stream policy optimization.

[2] Self-Evolving Curriculum for LLM Reasoning

[3] Can prompt difficulty be online predicted for accelerating rl finetuning of reasoning models?

[4] Online Difficulty Filtering for Reasoning Oriented Reinforcement Learning

**Questions:**

see weakness

---

### Official Review · Reviewer_GTCw · 2025-10-30

**Soundness:** 1
**Presentation:** 2
**Contribution:** 2
**Rating:** 4
**Confidence:** 4

**Summary:**

This paper aims to address the limitations of reinforcement learning fine-tuning for multimodal reasoning. The authors identify three main limitations in existing methods: Mixed-difficulty corpora, Flat reward schedules, and Absent difficulty awareness. To overcome these problems, they propose an RL framework that incorporates difficulty priors through offline data curation, online advantage differentiation, and difficulty hints. The approach uses a U-shaped accuracy distribution to filter overly easy or overly difficult samples, reweights training data based on group accuracy, and introduces difficulty hints to guide reasoning on harder problems. Experiments show that the method achieves competitive results on mathematical reasoning benchmarks (MathVista, MathVerse, and MathVision).

**Strengths:**

- The three proposed components directly correspond to the three stated limitations, providing a comprehensive and integrated solution.
- The proposed method achieves competitive performance with a small amount of training data (a total of 2.6K samples).

**Weaknesses:**

- In Limitation 2, the authors claim that flat reward schedules are limiting, but they provide no citations or theoretical support.
- In Limitation 3, the authors state that difficulty awareness is absent, but they offer no direct experiments or prior studies that provide dedicated empirical support for its causes.
- The data only come from the two math reasoning datasets. The analyses are unlikely to represent real-world task distributions.
- The replacement of GRPO std normalization with difficulty adaptive weighting is supported by empirical results. However, the theoretical justification is light and relies mainly on prior claims that std normalization induces difficulty bias, with no more detailed analysis.
- Steep Exponential is only described verbally without an explicit formula, which limits reproducibility.
- The experiments include a Random baseline, but the paper does not explain how the random strategy is implemented.
- The paper gathers external methods across SFT, RL, and SFT+RL, but it does not run a unified comparison on the same model with SFT, RL, and SFT + RL. As a result, it is hard to attribute the performance gains solely to the proposed method.
- The paper does not report the results of GRPO on the same model Qwen2.5-VL-7B-Instruct, and it does not compare against GRPO variants (e.g., DAPO[1], SRPO[2]).
[1] Yu Q, Zhang Z, Zhu R, et al. Dapo: An open-source LLM reinforcement learning system at scale[J]. arXiv preprint arXiv:2503.14476, 2025.
[2] Zhang X, Wang J, Cheng Z, et al. Srpo: A cross-domain implementation of large-scale reinforcement learning on LLM [J]. arXiv preprint arXiv:2504.14286, 2025.

**Questions:**

1. On which base model was the U-shaped accuracy distribution obtained? Would a similar distribution appear if a different model were used?
2. During inference, are difficulty hints still explicitly provided? If so, how is this data obtained?
3. Could you please provide the exact formula for the Steep Exponential function?
4. In your experimental results, you report “random” as a baseline. Could you clarify how this random baseline was implemented?
5. I would like to know how the proposed method compares with the standard GRPO and its variants (e.g. DAPO, SRPO) when trained on the same model.

---

### Official Review · Reviewer_XU3F · 2025-10-30

**Soundness:** 2
**Presentation:** 3
**Contribution:** 2
**Rating:** 4
**Confidence:** 3

**Summary:**

This paper uses "difficulty prior" to improve RL fine-tuning for multimodal inference: offline filtering of overly easy/overly difficult samples to focus on learnable intermediates; online non-linear reweighting of advantages based on in-group accuracy to strengthen gradients for challenging problems; and a second-stage approach that adds difficulty cues to more difficult samples to guide deeper thinking. With only 2.6K data points, it significantly improves multiple benchmarks and demonstrates good generalization and data efficiency.

**Strengths:**

Using 2.6K two-stage RL datasets, the authors outperform or match SFT or SFT+RL methods that utilize hundreds of thousands/millions of datasets on MathVista/MathVerse/MathVision, demonstrating outstanding cost-effectiveness.

The authors' online difficulty-weighted approach effectively amplifies gradients in all challenging problems and suppresses interference from easier problems, addressing the zero-gradient/gradient dilution issues of GRPO in scenarios with mixed difficulty levels.

The authors' use of difficulty hints makes the model more concise on easier problems and more in-depth on challenging ones.

**Weaknesses:**

The difficulty metric for offline algorithms depends on the base and sampling strategy: offline algorithms use the accuracy of multiple samplings on the base as the difficulty, which is strongly affected by the model, temperature, and number of samplings, making it difficult to apply this algorithm freely.

The benchmark testing is insufficient, lacking some key benchmarks such as MMMU, EMMA, and WeMath.

**Questions:**

Please see the weaknesses

---

### Note · Authors · 2025-11-14

I have read and agree with the venue's withdrawal policy on behalf of myself and my co-authors.